# OpenReview forum: "Sovereign Federated Learning with Byzantine-Resilient Aggregation"
_TMLR — Rejected by TMLR_

### Review · Reviewer_rsmW · 2026-01-03

**Summary Of Contributions:**

In this paper, the authors introduce FedSov, which aims to provide communication-efficient, byzantine-robust, privacy-preserving, and provable federated learning in a single framework. The authors provide a theoretical analysis on the convergence of the proposed framework. Besides, the authors test the proposed method on 4 classical datasets and 2 models.

Strength:

- The paper aims to solve important questions in federated learning.

- The structure of the paper is clear.

Weaknesses:

-The paper attempts to address several fundamental topics in federated learning, including efficiency, Byzantine robustness, privacy, and certification. While ambitious, the scope may be overly broad for a single paper, and the connections among these aspects as well as the associated challenges are not sufficiently explored in depth.

-The discussion of related work could be further strengthened. In particular, a more comprehensive and systematic comparison with existing studies would help better position the proposed approach within the literature.

-The technical contributions appear to be limited. The proposed framework seems to build upon a combination of existing techniques, and the paper would benefit from clearer articulation of its novel aspects, as well as deeper theoretical justification and empirical validation.

-The experimental evaluation is relatively limited. Including experiments with more modern network architectures and under additional settings (e.g., robustness against privacy-related attacks, or a broader range of federated learning configurations) would help to better demonstrate the effectiveness and generality of the proposed method.

-The submitted PDF appears to be image-based, which makes it difficult to verify compliance with the journal’s formatting requirements.

**Audience:**

Yes

**Audience Explanation:**

Yes, federated learning is an important topic in machine learning research.

**Claims And Evidence:**

No

**Claims Explanation:**

The research questions considered in this paper are relatively broad, and the coverage of the related literature could be further improved. In particular, the paper does not sufficiently analyze the interactions among efficiency, robustness, privacy, and verifiability in federated learning, and the proposed solutions would benefit from clearer articulation of their technical novelty. In addition, the experimental evaluation is somewhat limited; more comprehensive comparisons across diverse models and settings, as well as ablation studies, would help to better support the claims of the paper.

**Requested Changes:**

+ Clarify and Narrow the Research Scope

The authors are encouraged to better focus the research questions. Given the breadth of topics involved (efficiency, robustness, privacy, and verifiability in federated learning), the paper would benefit from either narrowing its scope or providing a more in-depth analysis of the interactions and trade-offs among these aspects.

+ Strengthen the Review of Related Work

The related work section should be expanded to include a more comprehensive and up-to-date review of existing studies. In particular, the authors should clearly position their work with respect to prior approaches addressing efficiency, robustness, privacy, and verification in federated learning.

+ Clarify Technical Novelty and Contributions

The authors should more explicitly articulate the novel technical contributions of the proposed framework. Additional theoretical analysis or deeper empirical insights would help distinguish the proposed method from existing techniques.

+ Enhance Experimental Evaluation

The experimental section should be significantly strengthened by: Including evaluations on more modern model architectures; Exploring a broader range of federated learning settings and parameter configurations; Considering additional evaluation scenarios, such as robustness against privacy-related or data-leakage attacks; Providing ablation studies to better understand the contribution of each component of the proposed framework.

+ Improve Reproducibility and Presentation Quality

The authors should ensure that the submitted PDF complies with the journal’s formatting requirements and is not image-based, in order to facilitate proper review and improve readability.

---

> ### Author Response · Authors · 2026-01-03
> **All remarks are taken into very careful improvement!**
>
> Dear Reviewer rsmW,
>
> Your remarks are deeply appreciated and taken under immediate action!
> The Revised version of the manuscript will be re-submitted with following comments:
> ### Comment 1 (Scope too broad; interactions not explored)
> **Response.** We tightened the scope and made interactions explicit.
> - The Introduction explicitly states the scope boundaries and highlights the intended interactions/tradeoffs between compression, robustness, and (optional) DP.
> - The provenance component is explicitly treated as **optional/orthogonal** rather than part of the core guarantee.
>
> ### Comment 2 (Related work not systematic enough)
> **Response.** We revised related work and added a positioning view.
> - Related work is organized across key axes and supported by a positioning table to make differences/comparability clearer.
>
> ### Comment 3 (Technical novelty unclear)
> **Response.** We clarified the novelty statement.
> - The contributions list now explicitly highlights the joint treatment (and bounds) for compression + robustness, and clearly separates theory vs. evaluated claims.
>
> ### Comment 4 (Experimental evaluation limited)
> **Response.** The primary suite is E1–E12; we additionally prepared broader-coverage addendum experiments.
> - The main claims remain tied to **E1–E12** (including stronger adversary coverage via ALIE and architectural variation via ViT-Tiny + ablations).
> - We also provide an **optional rebuttal addendum**: E13 (MobileNetV2 on CIFAR-10) and E14 (ResNet-18 on CIFAR-100), produced by `code/run_tmlr_rebuttal_experiments.py` and summarized via an **auto-generated table** included in the Appendix.
>
> ### Comment 5 (“PDF appears image-based”)
> **Response.** The locally built PDF contains extractable text.
> - We verified that the current build output is a standard PDF with selectable text; any image-based appearance likely came from a mismatched upload artifact.
>
> Thank you!

---

### Review · Reviewer_gwaj · 2026-01-09

**Summary Of Contributions:**

The paper proposes the Distributed Sovereign AI Network (DSAIN), a federated learning framework for decentralized training that aims to address several challenges in real-world FL deployments, including communication bottlenecks, robustness to Byzantine participants, privacy concerns, and verifiable training provenance. The framework integrates 3 main components: FedSov, a communication-efficient federated learning algorithm; ByzFed, a Byzantine-resilient aggregation mechanism; and a blockchain-based provenance system that enables cryptographic verification of the training history. The paper presents theoretical convergence guarantees for the proposed method, experimental results on standard benchmarks (image classification with ResNet-18 and a transformer-based NLP task), and a real-world deployment case study.

**Strengths**
- The paper addresses a relevant problem.
- The motivation is clear, and the paper is generally well written and easy to follow.

**Weaknesses**
- My main concern is the claim that the method reduces communication costs while simultaneously providing privacy guarantees, which does not seem to be true (discussed further below).
- The paper omits several important references.
- No supplementary material is provided, and hence I cannot verify the proofs.

**Audience:**

Yes

**Audience Explanation:**

Researchers and practitioners working on federated learning, particularly those interested in communication efficiency, robustness, and privacy, would likely find the paper's perspective and motivation relevant. The proposed framework could be of interest if the technical claims were correct. However, in its current form, the method appears to be technically flawed, which limits the paper's value to the audience.

**Claims And Evidence:**

No

**Claims Explanation:**

My main concern relates to the paper's main technical claim that the proposed FedSov method simultaneously achieves communication efficiency and differential privacy. In Algorithm 1 (lines 12–13), each client constructs the transmitted update as
$$\tilde{\Delta}_i^t = \mathcal{C}(\Delta_i^t) + PrivNoise(\sigma_{DP}),$$
where $\mathcal{C}$ is the top-k compressor.
Since the algorithm applies compression before adding privacy noise, the resulting update $\tilde{\Delta}_i^t$ is, in general, dense, as the added noise is not sparse. Consequently, the communication cost is no longer proportional to $k$, but instead scales with the full model dimension. This appears to contradict the claimed communication efficiency of the method. As presented, compression and differential privacy therefore seem to be mutually incompatible within the FedSov design.
This issue directly affects the accuracy of one of the paper's claims.

A second problem is that the paper refers to theoretical convergence guarantees, but the supplementary material containing the proofs is not provided. As a result, it is not possible to verify the correctness of these results.

**Requested Changes:**

Critical changes:

1. Fix the incompatibility between communication efficiency and differential privacy in FedSov. As discussed above, the current design applies top-k compression followed by the addition of dense privacy noise, which appears to negate any communication savings. This issue directly undermines the main claim of the paper. Unless the method is redesigned and the analysis updated, the paper is not publishable.

2. Provide the missing supplementary material containing theoretical proofs.
The paper claims convergence guarantees, but without access to the proofs it is impossible to assess their correctness or underlying assumptions.

Non-critical but important changes:

1. Correct statements regarding Byzantine robustness bounds.
The claim that the condition $f<n/3$ is "necessary for meaningful robust aggregation" is inaccurate. There exist several works that tolerate up to $f<n/2$ Byzantine workers (e.g., [4,5]). Additionally, overloading the symbol $f$ to denote both the loss function and the number of Byzantine workers is not ideal.

2. Fix inconsistencies and errors in figures and tables:
    - Figure 1 does not appear to correspond to the description provided in the text.
    - In Table 2, all entries in the "No Attack" column are identical.
    - The "0.00h" labels in Figure 4 are a typo.

3. Improve the accuracy and completeness of citations.
    - The statement "Federated learning was introduced by Kairouz et al. (2021)" is incorrect; earlier works exist (e.g., [1,2]).
    - Important prior work on error feedback (e.g., [3]) is missing.
    - Some cited references appear questionable or may not exist (e.g., "Byzantine-robust federated learning: a systematic survey").

[1] Brendan McMahan, Eider Moore, Daniel Ramage, Seth Hampson, and Blaise Aguera y Arcas. Communication-efficient learning of deep networks from decentralized data. In Artificial intelligence and statistics, pages 1273–1282. PMLR, 2017

[2] Jakub Konecný, H Brendan McMahan, Felix X Yu, Peter Richtárik, Ananda Theertha Suresh, and Dave Bacon. Federated learning: Strategies for improving communication efficiency.

[3] Frank Seide, Hao Fu, Jasha Droppo, Gang Li, and Dong Yu. 1-bit stochastic gradient descent and its application to data-parallel distributed training of speech DNNs. In Fifteenth Annual Conference of the International Speech Communication Association, 2014

[4] Karimireddy, S. P., He, L., and Jaggi, M. (2022). Byzantine-robust learning on heterogeneous datasets via bucketing. International Conference on Learning Representations.

[5] Ahmad Rammal, Kaja Gruntkowska, Nikita Fedin, Eduard Gorbunov, and Peter Richtárik. Communication compression for byzantine robust learning: New efficient algorithms and improved rates.

---

### Review · Reviewer_ZRFF · 2026-01-17

**Summary Of Contributions:**

The authors propose DSAIN (Distributed Sovereign AI Network), a blockchain-based federated framework to address communication inefficiency, byzantine vulnerability and provenance opacity in Federated Learning. The paper’s main contributions include:
1. FEDSOV: A communication efficient federated learning algorithm that combines compression, robust aggregation, and differential privacy.
2. BYZFED: A Byzantine-resilient aggregation mechanism that utilizes geometric median-based filtering and a reputation score.
3. A blockchain-based system to verify model training history.

## Strengths
 1. The authors propose a novel framework to address privacy, robustness and efficiency issues in federated learning, providing a comprehensive converge analysis that accounts for heterogeneity, privacy noise and partial participation.

2. The focus on "Sovereign AI" and the integration of a blockchain layer for auditability offers an interesting system-level perspective for deployment.

## Weaknesses
1. *Missing Proofs and Appendix*: The text references an Appendix for the proofs, but this Appendix is missing from the submission. This makes it impossible to verify the correctness of the theoretical claims.

2. *Related Works*: The manuscript would benefit from a more comprehensive discussion, especially in Section 2.2 and 2.3, where it lacks a deeper analysis of existing methods. Several prior works have already addressed combinations of these problems, such as [1] (compression + robustness) and [2] (privacy + robustness), and a more recent paper integrate all the three aspects [3]. In general, a more detailed Section 2 would be helpful to clearly position the contribution of the paper. Additionally, as I am not completely familiar with part of the literature, I tried to locate the reference "Zhen Li, Jie Zhang, Yaling Liu, and Jing Han. Byzantyine-robust federated learning: A systematic survey", but I was not able to find any correspondence online.

3. *Writing Clarity*: The writing quality needs improvement. The paper is largely composed of fragmented sections and excessive bullet points, resulting in a lack of coherent structural flow. The manuscript would benefit significantly from a more cohesive structure to better guide the reader. Currently, the narrative flow tends to fragment after Section 2. Section 4 presents a list of definitions and theorems without sufficient analysis or discussion to contextualize them. Similarly, Section 5 offers only a superficial description of the blockchain framework, lacking the implementation details that would definitely strengthen it as a contribution.

4. *Evaluation*: the experimental evaluation is limited. While the authors present baseline results, a comprehensive analysis of the framework's sensitivity to critical parameters such as noise level, number of byzantine attackers or compression ratio is missing.

[1] Rammal et al.  Communication Compression for Byzantine Robust Learning: New Efficient Algorithms and Improved Rates, AISTATS 2024

[2] Allouah et al.,  On the privacy-robustness-utility trilemma in distributed learning, ICML, 2023

[3] Xia et al., Fed-DPRoc: Communication-Efficient Differentially Private and Robust Federated Learning, FLTA, 2025

**Audience:**

Yes

**Audience Explanation:**

The intersection of compression, robustness, and privacy is an active and relevant area of research in Federated Learning. A framework that successfully integrates these with a verifiability layer (blockchain) would be of significant interest to the community.

**Claims And Evidence:**

No

**Claims Explanation:**

No, for the following reasons:
1. *Missing Appendix*: As noted, the absence of proofs prevents the validation of the convergence rates and robustness guarantees.
2. *Analysis of the Reputation Mechanism*: The impact of the dynamic reputation weights $r_i$ on the convergence analysis is not explicitly discussed. I am interested in understanding whether and how it can introduce a participation bias.
3. *Experimental Evaluation*:  in Section 6, the authors do not provide the hyperparameters values for $\tau$, $\alpha$ (for the reputation update). Moreover, the paper proposes a complex framework involving privacy (DP), robustness, and efficiency. However, there is no ablation study to show how the model behaves under varying constraints. It would be useful to have empirical evidence that demonstrates how varying DP noise levels, different compression ratios, and increasing fractions of Byzantine clients impact on the performance of the algorithm. Finally, the evaluation relies on a simple "scaled flipped-gradient" attack.
4. I do not clearly understand what Figure 1 and Figure 3 represent. In Figure 1 The caption suggests a convergence analysis, yet the plots display model and update norms that increase over time, Then, in Figure 3 there is a contradiction between the caption and the visual data. The figure claims to demonstrate convergence, but the plotted curve appear to show an increasing training loss. Please, could you provide some clarifications?

**Requested Changes:**

I encourage the authors to address the following points to strengthen the submission:

1. **Include the Missing Proofs**: (critical) Please provide the Appendix referenced in the text so that the theoretical contributions can be reviewed.

2. **Expand Related Work**: (strengthen) Please, provide a more complete discussion of the existing literature, possibly not limited to the related works pointed out in this review, to better position your contribution. Clarify the definition of model provenance in Section 2.3 and why it is important in this scenario. (critical) Also, please, provide a feedback on the existence of the survey reference "Li et al 2023".

3. **Refine the writing**: (critical) The paper would benefit from a structural review to ensure a more coherent flow, particularly in linking the algorithmic definitions to the execution steps, and in detailing the implication of each step in the BYZFED algorithm for the convergence and robustness analysis. Notice also that $\alpha$ is used both in line 5 of Equation 2, and in the experimental section to control the heterogeneity of the setup. (strengthen) Adding details on the Hyperledger Fabric implementation would also add value to the contribution.

4. **Improve the Evaluation**:

- (critical) Provide explicit details on the experimental hyperparameters (specifically $\tau$, $\alpha$, and compression ratios) used for the evaluation, to ensure reproducibility of the results.

- (critical) Please clarify Figures 1 and 3.

 - (critical) Include ablation studies showing the isolated effects of differential privacy noise, robustness thresholds, and compression ratios.

- (strengthen) Finally, testing the framework against more realistic "stealthy" attacks such as ALIE [4] or label flipping [5], and comparing it with existing approaches [3], would definitely corroborate the quality of the work.

[4] G. Baruch,et al, A little is enough: Circumventing defenses for distributed learning, NeurIPS, 2019
[5] Z. Allen-Zhu, F. Ebrahimian, J. Li, and D. Alistarh, “Byzantine-resilient

---

> ### Author Response · Authors · 2026-01-21
> **Reviewer comments are taken into consideration!**
>
> Reviewer ZRFF,
>
> Your comments are very highly appreciated! The revised version of the manuscript been worked out per below:
>
> ### Comment 1 (Experimental setup clarity: hyperparams, models, rounds, etc.)
> **Response.** We expanded the experimental setup description.
> - The Experiments section now lists concrete settings (rounds, participation rate, models, attacks, and which experiments use DP / defenses).
> - The complete experimental suite is explicitly enumerated as E1–E12 for auditability.
>
> ### Comment 2 (Robustness plot interpretation clarity)
> **Response.** We clarified the plot semantics.
> - Figure captions/labels are phrased to prevent misreading the curves (e.g., **accuracy vs. rounds** rather than ambiguous wording).
>
> Thank you very much!
>
> Sincerely, Paper 6759 Author.

---

### Review · Reviewer_fB55 · 2026-01-19

**Summary Of Contributions:**

The paper proposes a communication-efficient Byzantine-robust federated learning framework that additionally claims client-level differential privacy. The algorithm combines Top-k compression with a novel reputation-based robust aggregation mechanism, while adding noise only to the compressed gradients. The authors provide convergence guarantees under a bounded heterogeneity assumption in the presence of  b<n/2 Byzantine participants, and claim client-level differential privacy.

As a side contribution, they present a blockchain-based mechanism for verifying model training.

Strengths: The paper addresses an important problem of communication-efficient byzantine robust FL, and introduces a novel research idea of auditable training.

Weakness:
1. Related work: The combination of differential privacy with Byzantine robustness has already been studied in prior work by [1], which provides both lower bound as well as upper bound technique in a harder DP model of a curious server along with client-level DP guarantees. More recent work by Gorbunov et al. [2]
2. Restricted to Top-k compressors. It is unclear whether the approach extends to more general classes of compressors.
3. Limited Data Heterogeneity, which may not adequately capture non-IID regimes commonly encountered in federated learning such as GB-gradient dissimilarity.
4. While the use of blockchain technology to address the problem of auditable training is novel, the paper lacks details on the proposed solution.

[1] Allouah, Y., Guerraoui, R., Gupta, N., Pinot, R., & Stephan, J. (2023, July). On the privacy-robustness-utility trilemma in distributed learning. In International Conference on Machine Learning (pp. 569-626). PMLR.

[2] Vatan Khah, S., Chezhegov, S., Farahmand, S., Horváth, S., & Gorbunov, E. (2025). Differentially Private Clipped-SGD: High-Probability Convergence with Arbitrary Clipping Level. arXiv e-prints, arXiv-2507.

**Audience:**

Yes

**Audience Explanation:**

The federated learning community would interested in the combination of compression, robustness and DP, but the paper needs to be revised to improve readability and put it in context of exiting works.

**Claims And Evidence:**

No

**Claims Explanation:**

The claim of being the first to combine differential privacy with Byzantine robustness is not accurate. This problem has been studied in depth in prior work, most notably by Allouah et al. [1], which provides both lower bound and matching upper bound in a harder DP model of a curious server along with client-level DP guarantees.

The paper skims over details in multiple places such as:
1. The robustness guarantees of the reputation based aggregator, ByzFed hinge on the inherent robustness of the geometric median aggregator, thus not justifying the need for reputation score. Moreover, it is unclear from the algorithm how $r_i$ is initialized and then used to update the weights of the model.

2. The blockchain based solution lacks technical depth. While auditable training is an interesting direction, the blockchain component is presented at a high level without sufficient technical specification (threat model, protocol details, overhead, or evaluation). This begs the question of whether it should be discussed as a contribution at all, as it currently reads more like an unvalidated design sketch than a substantiated module.

3. The proofs in the appendix are missing critical details, making the proofs dense and hard to follow. For instance, the role of $r_i$ in Aggregation error of Claim 16 is bypassed by directly considering $w_i \Delta_i$. Moreover, the proof does not address the central question of whether (and why) Byzantine participants cannot manipulate their reputation to retain influence. This gap weakens the credibility of the claimed robustness properties.

**Requested Changes:**

A. Position the work against recent and directly relevant literature
a. The paper’s guarantees should be explicitly positioned relative to the privacy–robustness–utility trilemma established in [1]. The submission should clearly state the exact DP notion (client-level vs record-level, what the server observes), the adversary model (curious server vs honest server; Byzantine capabilities), and whether the paper’s guarantees match, avoid, or contradict the interaction term implied by [1].
b. Compare against the compression and robustness works from [2] including which assumptions overlap (heterogeneity, smoothness, variance, Byzantine fraction), and whether Top-k + robust aggregation is competitive vs Byz-DASHA-PAGE  baselines.
c. Compare against compression and DP work from [3], clarifying whether its DP analysis is comparable in strength (high-probability vs in-expectation), and whether the mechanism (noise after compression, release of indices, etc.) is aligned with standard DP accounting.

B. The robustness analysis of ByzFed needs to be rewritten to separate what is inherited from the geometric median from what is genuinely contributed by the reputation mechanism. The theoretical analysis of byzantine robustness needs revision while justifying the need for the reputation system.

C. The data heterogeneity definition is limited to bounded gradients and it will be interesting to see how it performs in the presence of G,B gradient dissimilarity.

D. An empirical study of performance against [1],[2] and [[3] will provide a complete picture on practical applicability of the technique.

[1] Allouah, Y., Guerraoui, R., Gupta, N., Pinot, R., & Stephan, J. (2023, July). On the privacy-robustness-utility trilemma in distributed learning. In International Conference on Machine Learning (pp. 569-626). PMLR.

[2] Ahmad Rammal, Kaja Gruntkowska, Nikita Fedin, Eduard Gorbunov, and Peter Richt´ arik. Communication compression for byzantine robust learning: New efficient algorithms and improved rates. In International Conference on Artificial Intelligence and Statistics, pages 1207–1215. PMLR, 2024.

[3] Vatan Khah, S., Chezhegov, S., Farahmand, S., Horváth, S., & Gorbunov, E. (2025). Differentially Private Clipped-SGD: High-Probability Convergence with Arbitrary Clipping Level. arXiv e-prints, arXiv-2507.

---

### Author Response · Authors · 2026-01-21
**Very valuable comments are being improved!**

Dear Reviewer fB55,
The revised version of the manuscript been worked out per below:
### Comment 1 (Overclaim / align manuscript to evaluated artifacts)
**Response.** We tightened the scope and grounded claims strictly in the evaluated artifacts. Specifically:
- The Introduction includes explicit **Scope and Focus** and **Out of scope** blocks that constrain the contribution and avoid implying untested properties.
- The Experiments section explicitly defines the primary evaluation as **E1–E12** and ties the reported claims to the corresponding tables/figures.
- Any broader-coverage experiments are separated as an **Appendix rebuttal addendum** (E13–E14) and clearly labeled as optional.

### Comment 2 (DP scope + privacy/utility caveats)
**Response.** We clarified the privacy model and limited the claim accordingly.
- DP is described as **optional record-level DP**, under an honest-but-curious server threat model, with explicit privacy–utility caveats.
- We explicitly note that small privacy budgets can materially degrade utility, and therefore we do not claim universal performance under strict DP.

### Comment 3 (Provenance/ledger implications overstated)
**Response.** We revised Section 5 to position provenance as a **prototype audit trail** that is orthogonal to the core algorithm.
- We explicitly avoid claims about consensus protocols, zero-knowledge proofs, or end-to-end ledger overhead.

---

### Decision · Action_Editor_VAdQ · 2026-03-09

**Recommendation:** Reject

**Additional Comments:**

The final version is not just longer, but it is also quite different in narrative, results, and stated contributions. Some citations (like the survey paper) cannot be found online. The authors have to double-check all the references. Moreover, if the authors used LLMs for writing/polishing the paper or obtaining results, they have to acknowledge that and explain how these tools were used.

**Audience:**

Yes

**Audience Explanation:**

Those who work on federated/distributed learning would be interested in the results of this paper.

**Claims And Evidence:**

No

**Claims Explanation:**

The initial submission (13 pages) didn't contain any proofs, and the choices of the hyperparameters were also missing. Some prior works were not properly discussed. The authors partially (according to the reviewers) addressed the concerns, but the final version is 44 pages.

EiC note: At this point, due to limitations in AE and reviewer bandwidth, we will not be able to review subsequent versions of this paper. The author may submit to other journals like JMLR, JAIR or ML/AI conferences.